# Evolution of the Crown Procyanidins’ Tetramer during Winemaking and Aging of Red Wine

**DOI:** 10.3390/foods11203194

**Published:** 2022-10-13

**Authors:** Alicia Jouin, Liming Zeng, Marina Riveiro Canosa, Pierre-Louis Teissedre, Michael Jourdes

**Affiliations:** Univ. Bordeaux, Bordeaux INP, Bordeaux Sciences Agro, INRAE, UMR 1366, OENO, ISVV, F-33140 Villenave d’Ornon, France

**Keywords:** crown procyanidins, condensed tannin, winemaking process, red wine aging, UPLC-UV-Q-TOF

## Abstract

Condensed tannins play a major role in the quality of red wine. After grape extraction, they quickly evolve thanks to different oxidation mechanisms. Recently, NMR identified a new sub class of condensed tannins, named crown procyanidins, in red wine. The crown procyanidins’ tetramer exhibits a macrocyclic structure composed of four (-)-epicatechin with an unusual cavity in the center of the molecule. These new tannins exposed a higher polarity than the linear tannins. In this work, the evolution kinetics of these crown procyanidins during the winemaking process and after aging of red wine in bottles were studied. Samples’ quantification was analyzed by UPLC-UV-Q-TOF. The concentration of cyclic and non-cyclic procyanidins was compared. During the winemaking process, crown procyanidins are mainly extracted at the beginning of the alcoholic fermentation and they remain stable until the end of the winemaking process. The high polarity and solubility of this new molecule in water was confirmed. During the aging of red wine in bottles, crown procyanidins’ concentrations are stable, whereas the non-cyclic tannins decrease dramatically. Finally, a strong oxygenation experiment confirmed the crown procyanidins’ resistance to oxidation and unique skills.

## 1. Introduction

Condensed tannins are one of the key factors in wine quality [1]. They have a significant impact on the organoleptic properties of the red wine, as well as in wine chemistry. In red wine, they are directly responsible for the colour [2] and taste [3,4,5]. Recently, a new subfamily of condensed tannin with an unusual skeleton has been reported and named crown procyanidin [6,7]. In red wine, crown procyanidin tetramer was detected by UPLC-UV-QTOF and the structure was characterized by NMR, which revealed a macrocyclic structure composed of four (-)-epicatechin monomers linked together via alternatively 4β→8 or 4β→6 B-type interflavanyl linkages (Figure 1). These new molecules exhibited very specific chromatographic characteristics compared with non-cyclic procyanidins. Crown procyanidins have a higher polarity, a better solubility than the non-cyclic procyanidins, and a resistance to phloroglucinolysis depolymerisation. Moreover, a recent study showed that these molecules are specifically located in grape skins and not in the other part of the grape berries. The unique skeleton of these new molecule’s families conferred it some unique properties in comparison with the non-cyclic procyanidins.

Solubility is one of the principal key factors during the winemaking process. It has been reported that 50% of the phenolic compounds are extracted during the winemaking process [8]. In fact, phenolic compounds are extracted during the first step of the winemaking process: the alcoholic fermentation and all the different existing maceration processes. Polyphenols, in particular condensed tannins, are extracted differently in terms of the function of their structure and of the medium [9]. On one hand, anthocyanins as well as low-molecular-weight flavanols are associated with a high solubility in aqueous medium [10]. Thus, they are extracted preferentially during the pre-fermentative maceration and during the beginning of alcoholic fermentation [11]. On the other hand, high-weight flavanols, flavonols, and phenolic acids are extracted during post-fermentative maceration in an alcoholic solution [12,13]. A long maceration period is sometimes used to extract more of this phenolic compound [14]. During the maceration step of the red wine making process, condensed tannins are extracted from the grape skins and seeds [15,16]. Their concentrations in red wine are influenced by grape varieties [17], as well as by the use of different technical processes. The type of maceration has a direct impact on the grape polyphenol extraction and especially condensed tannins [18,19]. In these winemaking processes, the time [20] and the temperature [21] are two major factors. Other maceration techniques were developed such as carbonic maceration [22]; cryomaceration [23]; or, recently, pulsed electric fields [24]. Anthocyanins and tannins are also extracted with aeration techniques such as pumping over or draining-off [25].

Wine aging is an important step in the future taste and quality of the red wine. During this period, phenolic compounds are involved in complex chemical reactions [26]. Condensed tannins undergo different structural polymerisation [27], transformation, or degradation. Direct condensation [28,29] or ethyl-bridge condensation [30,31] are one of the principal reaction in the condensed tannins’ evolution. The colour change is directly impacted by the anthocyanin decrease and evolution in polymerized pigments [32,33]. Condensation is possible between tannins and tannins [34] and tannins and anthocyanins [35], directly impacting the colour [36] and taste of the wine [37]. All of these chemical reactions influence the initial concentration of the condensed tannins and increase the polymers’ formation [26]. Moreover, some tannins’ intramolecular reaction has been detected in wine [38,39]. These oxidation reactions directly affect the tannins’ concentration [40,41]. The aims of this study were to determine the evolution kinetics of crown procyanidins from ripening to the aging of red wine. This study was separated into two parts. On one hand, crown procyanidins’ extraction and evolution during the winemaking process was carried out. On the other hand, the crown procyanidins’ evolution during againg of red wine in bottles was studied. During the work, crown tetramer tannins’ concentration was compared with the non-cyclic tetramer tannins’ concentration.

## 2. Materials and Methods

### 2.1. Chemicals

Deionized water was purified with a Milli-Q water system (Millipore, Bedford, MA, USA). Methanol (MeOH, analytical grade and HPLC grade) and ethanol (HPLC grade) were purchased from Prolabo-VWR (Fontenays/Bois, France). Water (Optimal^®^ LC/MS), MeOH (Optimal^®^ LC/MS) and formic acid (Optimal^®^ LC/MS) were obtained from Fisher Scientific (Geel, Belgium) for high-resolution mass spectrometry analysis.

### 2.2. UPLC-UV-Q-TOF

The UPLC-MS system used was an Agilent 1290 Infinity equipped with an ESI-Q-TOF-MS (Agilent 6530 Accurate Mass, Les Ulis, France). Chromatographic separation was carried out on an Eclipse Plus C18 column (2.1 × 100 mm, 1.8 µm). The solvents used were as follows: water with 0.1% formic acid for solvent A and methanol with 0.1% formic acid for solvent B at a flow rate of 0.3 mL/min. The gradient of solvent B for crown procyanidins methods was as follows: stay at 6% during 0.5 min; 6 to 95% in 13.5 min; stay at 95% during 4 min; and UPLC column was equilibrated for 3 min. For the non-cyclic procyanidins, other gradients were performed: 6% during 0.5 min, from 6% to 50% in 19.5 min, from 50% to 100% in 5 min, stay stable at 100% during 5 min, and then an equilibration for 3 min. ESI conditions were as follows: gas temperature and flow were 300 °C and 9 L/min, respectively; sheath gas temperature and flow were 350 °C and 11 L/min, respectively; and capillary voltage was 4000 V. The injection volume was 5 µL. The fragmentor was always set at 200 V. The quantification of the crown tetramer and non-cyclic B type tetramer was realised by Q-TOF. The collision energy used for the MS/MS fragmentation was set at 30. A standard sample containing 80 mg/L catechin was injected every eight samples in order to control the sensibility of the mass spectrometer.

### 2.3. Sample Quantification Methods

All of the winemaking, aging, and oxidized samples were collected and filtered at 0.45 µm. Samples were injected in triplicate in the UPLC-UV-Q-TOF. The data obtained were treated by MassHunter Qualitative Analysis software. For the crown procyanidins’ quantification, a linear calibration curve (2–65 mg/L) was established with pure crown procyanidins’ tetramer obtained from red wine with the protocol reported by Zeng et al., 2019 [6]. For the non-cyclic procyanidins, the concentration was determined with a calibration curve of catechin equivalent (2–100 mg/L). All of the compounds were quantified by MS. All of the compounds were integrated into extracted ion chromatogram. A fragmentation method in MS/MS was performed to confirm the structure of the crown and the non-cyclic procyanidins.

### 2.4. Wine Samples

#### 2.4.1. Samples Collected during the Winemaking Process

The winemaking samples were collected on a winery in Paulliac, near to Bordeaux. Cabernet Sauvignon grape was harvested at the end of September 2015 at maturity. On the same day, grapes were crushed and SO_2_ (4 g/hL) was added during the transfer in a stainless-steel tank (50 hL). Commercial yeast (522D, Lamothe Abiet, Canéjan France) and an extraction enzyme (Vinozym Vintage FCE, Lamothe Abiet) were added into the tank. The alcoholic fermentation was processed at 27 °C for 9 days. During this fermentation, two extraction treatments were used separately to improve the extraction of phenolic compounds: pumping over and draining-off. Pumping over is an extraction process used to quickly extract the phenolic compounds during the alcoholic fermentation. In this case, the liquid part of the tank is totally removed in another tank and the solid part stays at the bottom of the initial tank. Then, the liquid part (must) is pumping over the solid part. The draining-off is also an extraction process to extract the phenolic compounds at the beginning of the alcoholic fermentation. In this approach, a small part of the must is running off the tank and injected again on the top of the same tank to break the solid part of the must. Both extraction technique were used once a day during the 9 days of the alcoholic fermentation. For both extractions, a tank triplicate was used in the winery. In total, just during the alcoholic fermentation, 81 samples were collected A sample was collected every day before performing the extraction technique in a bottle. Finally, after the alcoholic fermentation, a maceration step of 10 days was carried out just before the free-run wine.

#### 2.4.2. Red Wines Samples Aged in Bottles

Two different red wine series were collected directly at the winery. The first one came from Languedoc-Roussillon, appellation Languedoc Montpeyroux, with the variety 100% Syrah. These series of red wine were composed of 16 wines with different vintages: 2011, 2010, 2009, 2008, 2007, 2006, 2003, 2002, 2001, 2000, 1999, 1998, 1994, 1993, 1992, and 1991. The second wine’s series originated from Bordeaux, appellation Pauillac, composed of 85% Cabernet Sauvignon, 12% Merlot, and 3% Cabernet Franc. These second wine’s series were composed of 17 wines with different vintages: 2016, 2014, 2012, 2010, 2008, 2006, 2004, 2002, 2000, 1998, 1996, 1994, 1992, 1988, 1986, 1984, and 1982. Each of the red wines were filtered on 0.45 µm before quantification in triplicate by UPLC-UV-Q-TOF.

#### 2.4.3. Samples from the Strong Oxidative Condition Experiment

To study the resistance of crown procyanidin tetramer to oxidation in the red wine, two commercials red wines from Bordeaux, made with only 100% Merlot, were used. These two wines have two different vintages, 2009 and 2016. The wine from 2009 was named “Old wine” and the wine from 2016 was named “Young wine”. At the beginning of the experiment, a bottle of both wines was characterized and the crown and non-cyclic procyanidins were determined by UPLC-UV-Q-TOF. The experiment was realised in triplicate. The temperature was controlled and keep at 20 °C during the experiment. Wine samples were oxygenated at 8 mg/L of oxygen once a week with a diffuser. Oxygenation was controlled with an oximeter (Oximeter HQ30D). The experiment was conducted over 15 weeks. A sample was collected once a week before the oxygenation and injected directly in the UPLC-UV-Q-TOF. The crown procyanidins and non-cyclic B type procyanidins were quantified with Mass Hunter.

### 2.5. Global Phenolic Compounds Analysis

Different analyses were performed to characterize the red wine as well as the principal phenolic compounds. Basic chemical wine analysis was characterized by FOSS WineScan 79,000 (FOSS, Nanterre). For the global phenolic compounds analysis, the total polyphenols index (TPI) as well as total tannins (TTs) were measured by spectrophotometric analysis. The anthocyanin’s concentration was realized by HPLC-UV analysis. For the TPI, red wines were diluted 100 times with water, then the optic density (OD) was measured at 280 nm using a quartz cell with a 1 cm diameter. The total polyphenols index was calculated as follows: ITP = OD × dilution. Each sample was measured three times and the means and standard variation were calculated. The total tannin estimated by the Bate–Smith reaction was measured by acidic hydrolysis of proanthocyanidin. Each red wine was diluted by 50 before the reaction. Then, in two different tubes, 1 mL of diluted wine was mixed with 500 µL of water and 1.5 mL of HCl 37%. One tube was heated at 100 °C in a waterbath for 30 min and the other one was cooled in ice for 30 min too. After, 250 μL of ethanol was added to both tubes and the absorbance difference was measured at 550 nm on a 1 cm optical path. The concentration was obtained by the following calculation:
C (g/L) = 50 × 0.3866 × ΔDO550

For the anthocyanins’ concentration, wines were filtered (0.45 μm) and directly injected. HPLC-UV analyses were performed by means of a Thermo Scientific Accela (Thermo Fisher Scientific, Waltham, MA, USA) with an Accela 600 pump module and a UV/Visible diode array detector and Xcalibur Software according to González-Centeno et al. (2017) [42]. Separation was performed on a reverse-phase C18 Nucleosil column (250 × 4.6 mm, 5 μm). The injected volume was 20 μL. The mobile phases were water/formic acid (95:5, *v*/*v*) (solvent A) and acetonitrile/formic acid (95:5, *v*/*v*) (solvent B), at a flow rate of 1 mL/min. Initial solvent B was set at 10%. The mobile phase gradient was as follows: 35% B at 25 min, 100% B at 35 min, 100% B from 35 to 40 min, 10% B at 41 min, and then 10% B for 4 min before the next injection. Eluting peaks were monitored at 520 nm. The mean peaks were identified by comparison with injected external standards and previous results. A calibration curve of malvidin-3-*O*-glucoside (5–50 mg/L, purity: 95%, Merck, France) was injected. Concentrations were expressed as milligrams of malvidin 3-*O*-glucoside equivalents per liter.

## 3. Results

### 3.1. Differentiation between Non-Cyclic Tetramer and Crown Procyanidins’ Tetramer in Red Wine

To determine the evolution and the extraction kinetics of the crown procyanidins’ tetramer and to compare it with a non-cyclic procyanidins’ tetramer, mass spectrometry techniques were used. On one hand, a specific method was created for the crown procyanidins. The injection of the pure crown tetramer standard confirm the MS molecule parameters (r.t. = 2.50 min, [C_60_H_48_O_24_ + H]^+^ = 1153.2608 Da). Moreover, the crown tetramer procyanidin concentration was determined by a crown tetramer procyanidin calibration curve injected on the UPLC-UV-Q-TOF. On the other hand, another method was developed for the non-cyclic procyanidins’ tetramer. The structure of the non-cyclic procyanidins’ tetramer (r.t. = 4.75 min, [C_60_H_50_O_24_ + H]^+^ = 1155.2765 Da) was confirmed by MS-MS. The concentration of this non-cyclic tetramer was determined by a (+)-catechin equivalent calibration curve. To confirm the MS data, Table 1 displays the difference between the masses calculated and the measured masses. The difference was under 5 ppm for both molecules, confirming the molecules’ formula.

### 3.2. Extraction Kinetics of the Crown Procyanidin Tetramer during Winemaking

The extraction kinetic of the crown procyanidins tetramer was determined during alcoholic fermentation and compared with the extraction of total anthocyanins and total condensed tannins. The crown procyanidins’ concentration was determined by UPLC-UV-Q-TOF, while the anthocyanin concentration was determined by HPLC/Vis at 520 nm and the total condensed tannins concentration by the Bate–Smith reaction. The extraction percent of different polyphenols and the sugar consumption during the time of the alcoholic fermentation is reported in Figure 2. During alcoholic fermentation, sugar is consumed by the yeast and directly transformed into alcohol, as reported in the purple curve in Figure 2. At the same time, most of the polyphenols are extracted from grape seed and skin. Regarding the extraction kinetic of anthocyanins and condensed tannin, as expected, the anthocyanins were extracted before the condensed tannins. Indeed, it is well known that anthocyanins are highly water soluble compounds, while condensed tannins need the presence of alcohol to be extracted from the seed and skin [16]. At the time, the extraction kinetics of the crown procyanidins tetramer was unexpected for a compound that belongs to the condensed tannins family. Indeed, as reported in Figure 2 (blue curve), the concentration of the crown procyanidins’ tetramer increased very quickly. After just one day of fermentation, there was already 40% of the final concentration of the crown procyanidins’ tetramer extracted, while at this point, the extraction levels of anthocyanins (red curve) and the other condensed tannins (green curve) were very low. Thus, the concentration of the crown procyanidins’ tetramer will keep increasing. Indeed, after three days of maceration, 68% of the final concentration of the crown procyanidins tetramer was already extracted, whereas only 60% of anthocyanins and 30% of the condensed tannins were extracted. The extraction differences could be explaining by a difference in the alcohol’s percentage in the media as well as by the hydrophilic differences between the different polyphenols extracted. At the beginning of the alcoholic fermentation, hydrophilic molecules like anthocyanins are more extracted than non-cyclic condensed tannins, which need alcohol to be extracted. Crown tannins are also highly hydrophilic molecules [6] and are extracted in water media in the absence of alcohol at the beginning of the alcoholic fermentation. Over all, the crown procyanidins’ tetramer has a behavior more similar to anthocyanins than to non-cyclic condensed tannins.

Following this first observation regarding the extraction kinetic of the crown procyanidins, the impact of the extraction process on the crown procyanidins concentration was studied. Two different extraction processes were compared: pumping over and draining-off. These two processes are used during alcoholic fermentation to oxygenate the media for the development of the yeast and to modulate the polyphenol extractions’ level. During the draining-off process, a small part of the must was pumped out of the tank and pumped back to the upper part, directly on the cap. The pressures of the falling juice modify the organization of the pomace cap and improve the extraction yield. In comparison with the draining-off, the pumping over process is a more aggressive extraction. In fact, the whole liquid part of the tank is totally removed in another container. At this moment, the cap stays on the bottom of the tank and all of the liquid is pumping over. This extraction techniques induces a full restructuration and immersion of the pomace cap. The extraction kinetics of the crown procyanidin tetramer during these different extraction processes was monitored by UPLC-UV-Q-TOF (Figure 3). The crown procyanidins’ tetramer concentration was different for the two different methods. For the draining-off methods, the concentration in crown procyanidin tetramer was stable in the beginning of the treatment and then increased after a few days. An opposite tendency was observed for the pumping over methods. An increase in the crown procyanidins’ tetramer at the beginning of the fermentation was measured and then they stabilized until the end. In fact, pumping over is a harsher method, which damages the integrity of the berries skin, where the crown procyanidins’ tetramer are located, faster. So, the pumping over technique extracted the crown procyanidins faster than the draining-off technique, which is, in the end, the more efficient technique. Finally, the crown procyanidin tetramer concentration was higher with the draining-off process, which promotes a continuous extraction of the phenolic compound. Moreover, a last sample was collected 11 days after the alcoholic fermentation and just before the separation between the grape pomace and the wine. The concentration of the crown procyanidin tetramer was quantified and it appeared that crown procyanidins remain stable from the last day of the alcoholic fermentation and 11 days after. This enological study proved the uncommon behavior of the crown procyanidins, different from the linear condensed tannins.

### 3.3. Evolution of Crown Procyanidins during Aging of Red Wine

After a first study during the winemaking process, the evolution of the crown procyanidin tetramer during the aging of red wine in bottles was studied. A comparison between the non-cyclic condensed tannins’ tetramers and the crown procyanidins’ tetramers was analyzed. First of all, two sets of wine samples were collected: one from Pauillac’s appelation with wine composed by 17 vintages from 1982 to 2016. This set of wine was created with a majority of Cabernet Sauvignon and a minority of Merlot and Petit Verdot. Then, a second set of wine from Montpeyroux’s appellation composed of 16 vintages between 1991 and 2011 was collected. This set of wine was exclusively composed of the Syrah variety. All of the wines of each set were elaborated in the same winery for each appelation with a similar process and were stored properly at the winery. The concentration of crown procyanidin tetramer and the main non-cyclic tetramer was quantified by UPLC-UV-Q-TOF for the two sets of wine. The concentrations of cyclic and non-cyclic tetramer as a function of the vintage are represented in Figure 4 for appellation Montpeyroux and for appellation Pauillac. For both figures, the non-cyclic tetramer was represented in black and crown tetramer was represented in blue. Total polyphenolic richness (TPI) was measured in grey. Each representation was ordered from the youngest to the oldest wine. For the Montpeyroux wines, the non-cyclic procyanidin tetramer concentration decreased during the aging of red wine in bottles from 50 mg/L to less than 10 mg/L. At the same time, the crown procyanidin tetramer concentration was quantified between 25 and 45 mg/L over all vintages. Despite the variation in concentration between vintages, the concentration of crown procyanidin tetramer showed a rather stable tendency over time. Indeed, the crown procyanidin tetramer appears to be more influenced by the vintage than by aging. Moreover, the total polyphenolic concentration (TPI) has a similar tendency to the crown procyanidins’ concentration. TPI was more impacted by the vintage than by the wine aging. On the opposite side, the non-cyclic procyanidin tetramer decreases constantly over time. This last result proved the impact of the vintage and aging on the non-cyclic tannins. This study continued with the Pauillac wines set. For the wine samples from Paulliac, the non-cyclic procyanidin tetramer had a regular decrease during aging of red wine from 48 mg/L to less than 15 mg/L, while at the same time, the crown procyanidin tetramer concentration stayed stable, ranging between 32 and 56 mg/L. During the aging time, the evolutions of the cyclic and non-cyclic tetramer were totally different. While the crown tetramer remained stable, the concentration of the non-cyclic tetramer decreased. As reported in Figure 4, the concentrations of the two compounds were similar for the youngest vintage, 2016. In fact, the concentration of the non-cyclic tannins was divided by three for the 1982 vintage, whereas the concentration of crown procyanidin tetramer remained the same in 2016 and in 1982.

Overall, the evolution of the crown procyanidin tetramer and the non-cyclic procyanidin tetramer concentration shows similar trends between the two sets of wines, even if the sets of wines are created with different grape varieties and in different regions. Finally, the kinetic evolution of the crown procyanidin tetramer was mainly stable during the aging of red wine in comparison with the non-cyclic tetramer, which exhibits a drastic decrease with the aging time. In fact, the concentration of the crown procyanidin tetramer appeared to be more influenced by the vintage as well as total polyphenolic richness than the wine aging. Such behavior could indicate that the crown procyanidins’ tetramer is more resistant against oxidation than the non-cyclic condensed tannins. Indeed, according to its unusual macrocyclic structure, the crown procyanidin tetramer does not have any first unit or terminal unit of the polymeric chain, which are known to be involved in the oxidation and evolution mechanism of the condensed tannins [32,39]. Once again, as observed with the extraction kinetics, the crown procyanidin tetramer exhibits a different evolution pattern compared with the non-cyclic condensed tannins.

### 3.4. Crown Procyanidins under Strong Oxidative Condition

To confirm the possible resistance of the crown procyanidins tetramer to oxidation, rwo red wine was submitted to strong oxidative experiments. The two wines were composed of 100% Merlot; one was two years old and the other one nine years old. The evolution of the crown procyanidins under strong oxygenation conditions in two different matrices was studied. Each sample was submitted to an addition of 8 mg/L oxygen once a week for 20 weeks. Every week, the evolution of the crown procyanidins’ tetramer concentration as well as the non-cyclic procyanidin tetramer concentration was measured by UPLC-UV-QTOF. Concentrations of cyclic and non-cyclic tetramer tannins are presented in Figure 5. At the beginning of the study, the concentration of both tetramers in the older wine was measured at 51 mg/L for crown procyanidin tetramer and 47 mg/L for the non-cyclic procyanidin tetramer. Week after week, the decrease in the non-cyclic tetramer was quantified to reach only 14 mg/L after 20 weeks. On the opposite side, the crown procyanidin tetramer concentration stayed stable over the 20 weeks. A similar trend was observed with the young red wine. The concentration of the non-cyclic procyanidin tetramer was measured at 71 mg/L at the beginning of the experiment, which was higher than the crown procyanidin tetramer concentration at 35 mg/L. However, after 20 weeks, both tetramers exhibited a similar concentration around 30 mg/L. These last results proved the drastic decrease for the non-cyclic procyanidin tetramer, whereas in the same oxidative conditions, the concentration of crown procyanidins’ tetramer remained stable. In fact, the concentration of crown procyanidins’ tetramer remained stable during the 20 weeks, whereas the concentration of the non-cyclic procyanidin tetramer dropped. The absence of an initial and terminal unit in the crown procyanidins’ structure explains the specific oxidation’s resistance. In fact, in organic chemistry, the cyclic structure is chemically really stable and only a strong reaction can break the cycle. In wine, an oxidative reaction always starts at the initial or terminal unit of the condensed tannins [32,38]. The crown procyanidins did not have any initial or terminal unit. So, they are more stable and less impacted by chemical reaction.

## 4. Conclusions

In conclusion, the evolution of crown procyanidins during the winemaking process and during the aging of red wine was studied. The concentration of cyclic and non-cyclic tannins was compared. During the alcoholic fermentation, the crown procyanidins were mainly extracted at the beginning of the fermentation like the anthocyanins. These results are totally in opposition to the non-cyclic tannins’ extraction tendency. In fact, crown procyanidins are highly soluble in water and do not need ethanol to be extracted from the grape skin during the wine making process. Indeed, their extraction kinetic during the red wine making process appeared to be closer to the anthocyanins than the non-cyclic condensed tannins. An extraction process also proved the hydrosolubility of the crown tannins as well as their stability until the end of the fermentation. Then, the quantification of crown procyanidins during aging of red wine was studied. Crown procyanidins appear to be more resistant to oxidation than non-cyclic condensed tannins. During the aging of red wine in bottles, their concentration remained stable compared with the drastic decease in the non-cyclic procyanidins. These results were proven for two different sets of wine in different appellations in France. Finally, during a last oxidation experiment, crown tannins’ tetramer exhibited a strong resistance to oxidation and stayed stable against oxygen. At the same time, the non-cyclic procyanidins decreased dramatically until they were not quantifiable anymore. The specific cyclic structure of these new condensed tannins confers them all of these different properties and can represent a new future for the quality of wine.

## Figures and Tables

**Figure 1 foods-11-03194-f001:**
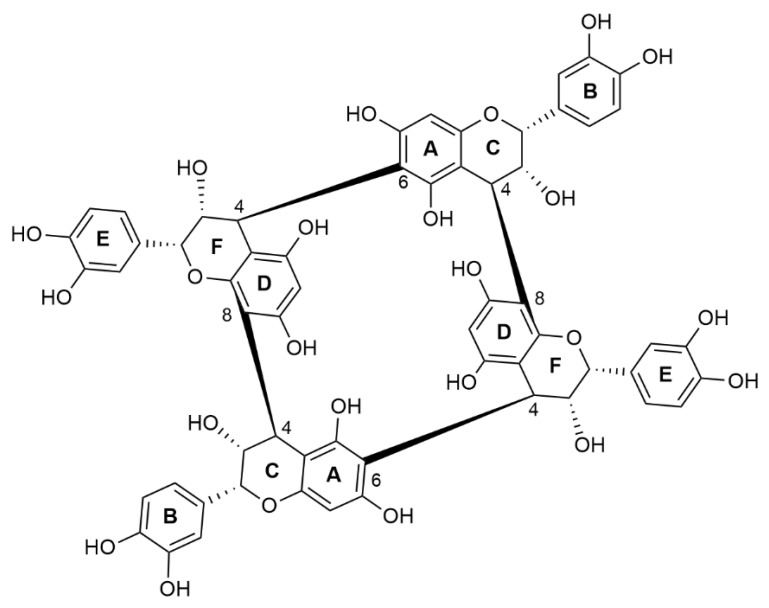
Structure of the crown procyanidin tetramer.

**Figure 2 foods-11-03194-f002:**
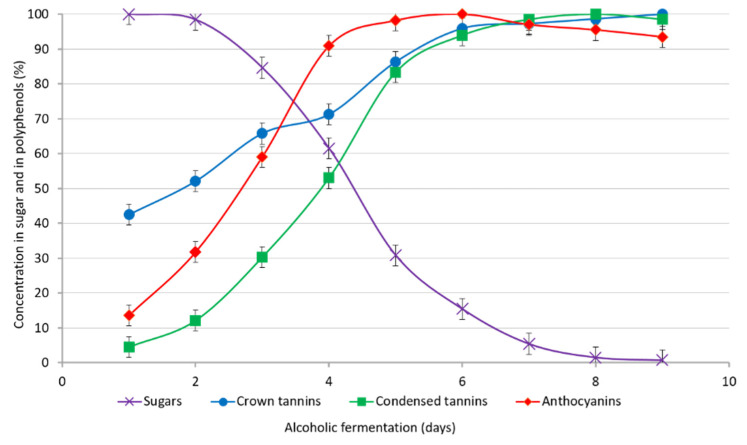
Evolution of sugar (in purple), anthocyanins (in red), condensed tannins (in green), and crown procyanidins’ tetramer (in blue) during the alcoholic fermentation expressed as a percentage according to their maximum concentration.

**Figure 3 foods-11-03194-f003:**
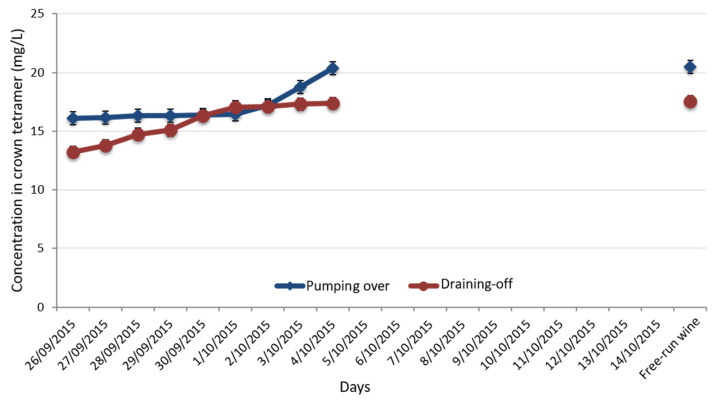
Impact of the extraction process on the crown procyanidins’ tetramer concentration.

**Figure 4 foods-11-03194-f004:**
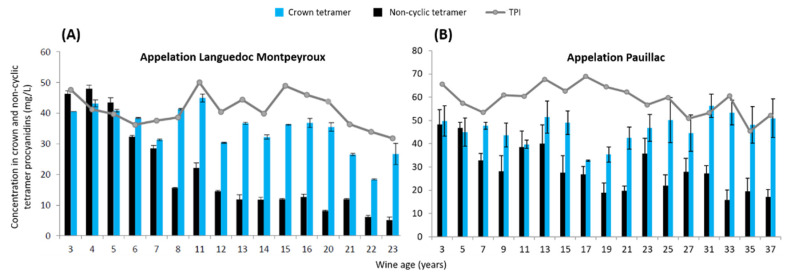
Concentration in the crown and non-cyclic procyanidins’ tetramer, as well as TPI during the aging of red wine in bottles. (**A**) Red wine for Languedoc Montpeyroux appellation and (**B**) red wine from Pauillac appellation.

**Figure 5 foods-11-03194-f005:**
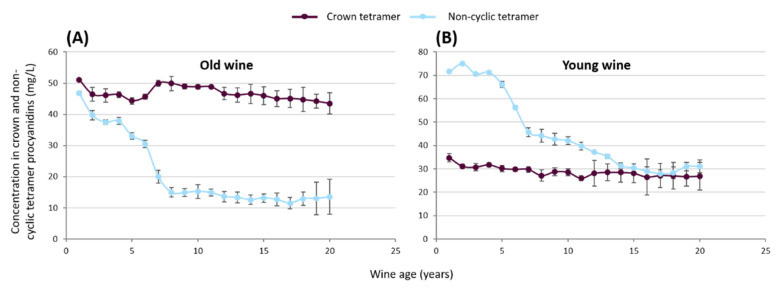
Evolution kinetics of crown and non-cyclic procyanidins’ tetramer in red wine under strong oxidative condition. (**A**) Old wine and (**B**) young wine.

**Table 1 foods-11-03194-t001:** Positive ions of B type procyanidins’ tetramer and crown procyanidins’ tetramer.

Compounds	Formula	Calculated *m/z*	Measured *m/z*	Diff (ppm)
B type tetramer	[C_60_H_50_O_24_+H]^+^	1155.2765	1155.2724	−3.55
	[C_60_H_50_O_24_+Na]^+^	1177.2584	1177.2555	−2.46
Crown tetramer	[C_60_H_48_O_24_+H]^+^	1153.2608	1153.2601	−0.61
	[C_60_H_48_O_24_+Na]^+^	1175.2428	1175.2428	0.00
	[C_60_H_48_O_24_+K]^+^	1191.2167	1191.2151	−1.34

## Data Availability

All related data and methods are presented in this paper. Additional inquiries should be addressed to the corresponding author.

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
