# Peer review of "Evolution of the Crown Procyanidins’ Tetramer during Winemaking and Aging of Red Wine"

_foods, 2022, doi:10.3390/foods11203194_

Round 1
Reviewer 1 Report
Dear Authors,
my compliments for your interesting study on kinetics evolution of crown procyanidins during winemaking and aging.
The results are clearly presented.
In my opinion just some moderate English changes are required:
for example:
1. non-cyclic instead of none-cyclic
2. lines 69-74 impact is repeated several times
3. technique not technic
4. lines 124-134 technique is repeated several times also. Please find a synonym.
Regards
Author Response
Reviewer 1
Dear Authors,
my compliments for your interesting study on kinetics evolution of crown procyanidins during winemaking and aging.
The results are clearly presented.
In my opinion just some moderate English changes are required: for example:
-non-cyclic instead of none-cyclic
R: In the text, none-cyclic have been changed for non-cyclic.
-lines 69-74 impact is repeated several times
R: In the text, the verb “impact” have been transformed by “affect” or “influence”.
-technique not technic
R: In the text, technic have been changed by technique.
-lines 124-134 technique is repeated several times also. Please find a synonym.
R: In the text, the word “technique” have been transformed by “process” or just removed.
Reviewer 2 Report
This is an interesting paper investigated the behavior of crown procyanidins tetramer during winemaking and aging of red wine. Please consider the following minor revisions:
Lines 38-40. Please provide more references.
Line 95. “0.5min” “0.4min” add space
Lines 97-98 “0.5min” “19.5min” “5min” “3min” add space also in other parts of the text
Paragraph 2.2. please specify the injection volume
Line 112. Please furnish more detail how the quantification takes place, integrating the area of a specific ion (quantitative ion) or on the TIC (total ions current)?
Line 119. SO2 (4 g/hL)
Line 186. Specify which were the external standard, their purity, supplier, and concentration solutions
Line 195, 199,. the number of atoms should be put in subscript, while the charge in superscript. Like in table 1.
Verify throughout the text that m/z must be in italic
Figure 2. Are there any error bars?
Figure 3. What does “ecoulage” mean? Should those points be connected to the previous ones?
Author Response
Reviewer 2
This is an interesting paper investigated the behavior of crown procyanidins tetramer during winemaking and aging of red wine. Please consider the following minor revisions:
-Lines 38-40. Please provide more references.
R: In line 38-40, an article on the crown procyanidins have been added.
-Line 95. “0.5min” “0.4min” add space
R: Spaces have been added in the text between the numbers and “min”.
-Lines 97-98 “0.5min” “19.5min” “5min” “3min” add space also in other parts of the text
R: Spaces have been added in the text between the numbers and “min”.
-Paragraph 2.2. please specify the injection volume
R: The injection volume was 5µL and have been added in the text.
-Line 112. Please furnish more detail how the quantification takes place, integrating the area of a specific ion (quantitative ion) or on the TIC (total ions current)?
R: Quantification have been detailed and was integrated by EIC (Extract ION Chromatogram).
-Line 119. SO2 (4 g/hL)
R: SO2 was corrected in the text by SO2.
-Line 186. Specify which were the external standard, their purity, supplier, and concentration solutions
R: For the external standard, malvidin-3-O-glucoside with 95% of purity from MERK was injected as a calibration curve from 5-50 mg/L.
-Line 195, 199, the number of atoms should be put in subscript, while the charge in superscript. Like in table 1.
R: In subscript, the number of atoms have been corrected like in Table 1.
-Verify throughout the text that m/z must be in italic
R: The all manuscript have been verified to make sure that all m/z are in italic.
-Figure 2. Are there any error bars?
R: Error bars was added to the initial Figure 2.
-Figure 3. What does “ecoulage” mean? Should those points be connected to the previous ones?
R: “Ecoulage” have been changed by “Free-run wine”. Those points don’t need to be connected because all the point connected correspond to the alcoholic fermentation and the last point at free-run wine was only collected by curiosity.